# Over-the-Counter Medication Use among Parents in Saudi Arabia

**DOI:** 10.3390/ijerph20021193

**Published:** 2023-01-10

**Authors:** Ola A. Mirdad, Ghada E. Esheba, Ahmed H. Mousa, Houriah Y. Nukaly, Yosra Z. Alhindi, Nahla H. Hariri, Nahla Ayoub, Safaa M. Alsanosi

**Affiliations:** 1Department of Pharmacology and Toxicology, Faculty of Medicine, Umm Al-Qura University, Makkah 24375, Saudi Arabia; 2Faculty of Medicine, Umm Al-Qura University, Makkah 24375, Saudi Arabia; 3Faculty of Medicine, Tanta University, Tanta 31111, Egypt; 4College of Medicine and Surgery, Batterjee Medical College, Jeddah 21442, Saudi Arabia; 5Department of Community Medicine and Health Care for Pilgrims, Faculty of Medicine, Umm AlQura University, Makkah 24375, Saudi Arabia; 6Institute of Cardiovascular and Metabolic Health, University of Glasgow, Glasgow G12 8QQ, UK

**Keywords:** self-medication, over the counter, parents, self-prescribing

## Abstract

Introduction: Self-medication is a growing public health concern worldwide. Studies have shown a gap between best practice and the current practice of using over-the-counter (OTC) medications. Despite being a well-recognised problem in Saudi Arabia, few studies have investigated OTC medication use in Saudi Arabia. Therefore, this study aimed to investigate the attitudes and knowledge of parents regarding OTC medication use in the Jeddah region, Saudi Arabia. Method: A cross-sectional study was carried out via an electronic questionnaire sent randomly to parents over four months, from 1 January to 30 April 2022. The participants’ characteristics and categorical variables were represented descriptively by frequency and percentage. A Chi-square test was used to test the relationship between the variables. Results: In total, 211 questionnaires were included in this study. Females represented 54.5% of the participants included in the study. Parents belonging to the 18-to-30-year-old group comprised the highest percentage (37.9%), and most of the parents (72.9%) had received an undergraduate education. Family physicians were the most common source (37.3%) of information about OTC medications, whereas more than half of parents purchased OTC medications from the community pharmacy (58.8%). While almost half of the parents (52.1%) visited a family physician when side effects of OTC medications appeared in their children, only (33.6%) stopped giving their children the OTC medicine. The relationship between the sociodemographic characteristics (including educational level, marital status, and employment status) and OTC drug consumption was significant (*p* < 0.001). Conclusion: Educational campaigns are needed to guide patients about the proper use of OTC medications. Studies on OTC medication use are lacking in Saudi Arabia in terms of its frequency, reasons for use, type of self-medication, and contributing factors.

## 1. Introduction

The self-medicating practice of using over-the-counter (OTC) medications is more common than prescription drug use and it is a growing public health concern worldwide [1]. Self-medication is defined as “the taking of drugs, herbs or home remedies on one’s initiative, or on the advice of another person, without consulting a doctor” [2]. OTC medications refer to all drugs that can be legally purchased without requiring a prescription from a registered medical practitioner [3]. OTC medications are primarily used to treat conditions that do not need direct medical attention or supervision [4].

Although regulatory authorities carefully propose OTC medications to encourage safe and effective use, studies have shown a gap between best practice and the current practice of using OTC medications and have also highlighted the need for education efforts to change this behaviour [5,6]. Despite the presence of several regulations for OTC consumption in Saudi Arabia, up to 81% of the general population has reported using OTC medication at some point in their life [7]. For instance, in the Dammam region, the results showed that more than half (66%) had insufficient knowledge about OTC medications, and 54% had a positive attitude towards self-medication [8]. In the Qassim region, there was a high prevalence of OTC medicine purchases from community pharmacies (75%), and the most common reason for buying OTC medicines was the repetition of a previous prescription by a healthcare provider (30%) [9].

Children represent more than 30% of the Saudi population and they are vulnerable to a variety of diseases [10]. Moreover, parents might use OTC medicines based on their earlier experiences, buy medicines based on a close relative’s advice, or buy them directly from the pharmacy without consulting a doctor [1]. Studies have shown that when children suffer minor symptoms, most parents’ initial reaction is self-medication, which might endanger the child’s life since parents generally lack sufficient medical knowledge and understanding, such as the possibility of OTC medications interacting with other medications, supplements, food, and drinks [11,12,13]. In addition, parents usually get health advice from family members, the media, friends, or other sources, which may influence their medical decisions on behalf of their children [14,15].

Inappropriate use of OTC medications has become increasingly clear to clinicians and healthcare systems in Saudi Arabia as irrational self-medication has several negative health impacts, such as drug resistance and interactions, adverse drug reactions, and polypharmacy [16]. Despite being a well-recognised problem, few studies have investigated OTC medication use in Saudi Arabia. Therefore, this study aimed to investigate the attitudes and knowledge of parents regarding OTC medication use in the Jeddah region, Saudi Arabia.

## 2. Materials and Methods

### 2.1. Study Design

A cross-sectional study was conducted among parents in the Jeddah region, Saudi Arabia. An electronic questionnaire was randomly sent to parents over four months, from 1 January to 30 April 2022. The participants were informed about the aim of the study, and the decision to participate in the study was voluntary and free.

### 2.2. Questionnaire Tool

The questionnaire was adapted from an earlier study by Orayj et al. [16]. It was designed in English and translated into Arabic, the local spoken language, by proficient speakers of both languages, and reviewed to suit the general population. It was divided into three main parts: The first part included sociodemographic information. The second part was about sources of information, obtainment, and dosing of OTC medications. The third part presented the side effects of OTC medications and parents’ responses to those side effects.

### 2.3. Study Populations

We included parents above 18 years of age who were using OTC medications for their children and live in the Jeddah region, Saudi Arabia. Participants who were unable to provide informed consent and parents of children with serious medical or surgical conditions requiring hospitalisation were excluded from the study.

### 2.4. Sample Size and Data Collection

The sample size was calculated using Slovin’s formula, with a population size of 109 participants from a recently published study in the Qassim region, Saudi Arabia by AlGhofaili et al., 2021, with a confidence interval (CI) of 95% and a margin of error of 5% [9]. The participants were randomly approached by sending the electronic questionnaire through social media (including Twitter and WhatsApp) and Google Forms, and the responses were downloaded. Participants who did not answer all 13 questions from the three-part questionnaire were excluded. The data were collected from the spreadsheets provided by Google Forms and transferred to Microsoft Excel.

### 2.5. Statistical Analysis

Data were analysed using SPSS version 23.0 (SPSS Inc., Chicago, IL, USA). Variables were presented as frequency and percentage. A Pearson Chi-square test was used to measure any differences. A *p*-value of <0.005 was considered statistically significant.

## 3. Results

A total of 250 questionnaires were collected, of which 39 were excluded because of incomplete responses, giving a response rate of 84%. In total, 211 questionnaires were included in this study. Females represented 54.5% of the participants in the study, and Saudis represented most of the participants (90.5%), as shown in Table 1. In terms of age groups, parents belonging to the 18-to-30-year-old group comprised the highest percentage (37.9%), followed by the 31-to-40-year-old group (34.6%) and the 41-to-50-year-old group (17.1%). Most of the parents (72.9%) had received an undergraduate education, 86.7% of the participants were married, 60% were employed, and almost half of the parents had a monthly income of more than SAR 10,000.

There was a dependent relationship between sociodemographic characteristics (including educational level, marital status, and employment status) and OTC drug consumption in children (*p* < 0.001), as shown in Table 2. However, sociodemographic characteristics (including gender, nationality, and age) were not related to OTC drug consumption in children (*p* = 0.027, *p* = 0.025, and *p* = 0.027, respectively).

As for sources of information, family physicians were the most common source (37.4%) of information about OTC medications, followed by community pharmacists (32%), the internet/social media (15.5%), and friends/family (15.1%), as shown in Figure 1. Concerning sources of obtainment, more than half of parents purchased OTC medications from the community pharmacy (58.8%), followed by hospitals (34.6%), leftovers from previous prescriptions (5.2%), and finally from retailers/supermarkets (1.4%). To calculate the dosage of OTC medications, 44% of the parents ask the community pharmacist, 40.3% read the package leaflet, 10% decide the dosage depending on the severity of the symptoms, and 5.7% rely on their previous experience with OTC medications.

As shown in Table 3, the most reported side effect was drowsiness (26.5%), followed by loss of appetite (21.3%), runny nose (20.3%), itchy skin (17%), and diarrhoea (16.5%). As to parents’ response when side effects of OTC medications appear in their children, almost half of the parents (52.1%) visit a family physician, 33.6% stop giving the OTC medicine, and only 10% use another OTC medicine.

As shown in Figure 2, the most common OTC medications used by parents for their children included acetaminophen (pain reliever, 93.3%), followed by ibuprofen (non-steroidal anti-inflammatory drug, 37.4%), dimetindene maleate (antihistamine, 30.8%), chlorpheniramine maleate (antihistamine) and xylometazoline hydrochloride (nasal drop decongestant), which had the same percentage (20.5%), loratadine and desloratadine (anti-allergic), which also had the same percentage (20.3%), pseudoephedrine (decongestant, 16.1%), dextromethorphan (cough suppressant, 15.6%), salbutamol (bronchodilator, 14.7%), and fexofenadine (anti-allergic, 10.9%).

## 4. Discussion

Using OTC medications without any consultation with a medical professional may result in wrong self-diagnosis and treatment, failure to reach an appropriate healthcare facility, inadequate dose, and prolonged treatment, besides wasting the resources of the country [1]. The main aim of this study was to investigate the attitudes and knowledge of parents regarding OTC medication use in the Jeddah region, Saudi Arabia. Irresponsible self-medication is common in Saudi Arabia, given that sources of information and knowledge and perception of self-medication are insufficient [17].

As for education, there was a clear trend towards more OTC medication use among parents with an undergraduate education. Similarly, two clinical studies showed that almost all of the students (99.7% and 92.3%), including undergraduates and postgraduates, had a positive attitude towards the use of OTC medications, and only a few students used OTC medicine inappropriately, such as not reading the instructions or taking more than the recommended dose [18,19]. We also found an explicit association between education and self-medication with OTCs (*p* < 0.001). This result is consistent with earlier studies [20,21,22,23], which found that individuals with higher educational levels tend to have better health knowledge about diseases and medications, higher self-confidence in making proper decisions about self-medication, and less faith in the quality of formal health services; hence, they are more likely to use OTC medications.

In our study, parents with a high income (a monthly income of more than SAR 10,000) represented the highest percentage of parents using OTC medication. We also found a dependent relation (*p* < 0.001) between marital status and self-medication with OTCs and between employment status and self-medication with OTCs. Several studies conducted in high-income countries have concluded that marriage is positively related to high income, and those with a higher income have a greater tendency to use OTC medications [24,25]. For example, a clinical study conducted in Denmark showed that individuals with a low income tend to use prescribed medication more, whereas those with a high income tend to use more OTC medications [26]. Another study, conducted in Austria, showed that a high income was linked to the use of OTC medications and the less educated were less likely to take OTC medicines (which is consistent with our results regarding educational level) [27]. People with a high income could have greater access to high-quality OTC medications, more financial flexibility, and less economic pressure when taking days off to get better [28,29].

The community pharmacy is a rapidly growing sector in Saudi Arabia and has lately been given more attention by the government [30]. In our study, more than half of the OTC medications that parents bought were from community pharmacies. Furthermore, more than 40% of parents relied on information received from community pharmacists about the appropriate dosage of the OTC medication. However, low adherence to evidence-based practices by community pharmacists in Saudi Arabia when dispensing OTC medication has been reported, as only 40% of pharmacists dispensed OTC medications according to the recommended guidelines [31,32]. Those results highlight the importance of monitoring community pharmacists and encouraging them to keep updated while taking the initiative to actively guide patients and provide information about the proper use of OTC medications and their possible negative outcomes when dispensing medications [33].

In general, parents use OTC medications every so often to treat minor ailments such as fever, toothache, pain, or the common cold [34]. The reason parents might act this way is to provide their child with relief as fast as possible without having to spend hours in the emergency room or a paediatric clinic [35,36]. In our study, acetaminophen was the most commonly used OTC medication as most parents are alarmed when their child has a fever and want to reduce it as soon as they can [37]. Similarly, evidence from multiple research projects has shown that acetaminophen is the first choice for parents to treat fever or minor pain [38,39,40]. Although acetaminophen is the drug of choice in managing illnesses at home, the risk of overdose or liver toxicity cannot be ignored [30].

The current study was limited by the study design as data were obtained by self-reporting; hence, the responses to the questionnaire may not reflect the actual attitudes or behaviours of the parents. Another limitation is that we did not capture the responses of other parents who do not use social media. Despite these limitations, to our knowledge, the present study is a novel contribution to the literature on the use of OTC medications by parents in the Jeddah region, Saudi Arabia.

## 5. Conclusions

There was a high prevalence of OTC medication inquiries and purchases from community pharmacies in the study. Educational campaigns for responsible self-medication should be strengthened through encouraging community pharmacies to take the initiative to actively guide patients and educate them about the proper use of OTC medications. Furthermore, additional research is needed to monitor OTC use among parents and identify factors contributing to self-medication to provide successful strategies to improve medication practices in Saudi Arabia.

## Figures and Tables

**Figure 1 ijerph-20-01193-f001:**
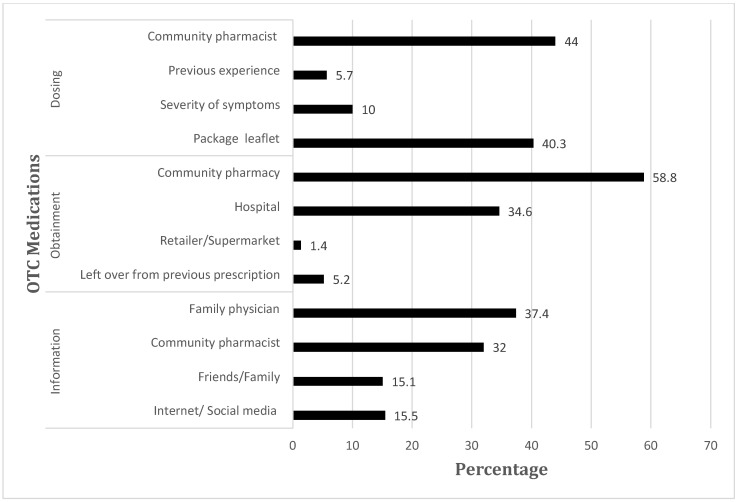
Sources of information, obtainment, and dosage of OTC medications.

**Figure 2 ijerph-20-01193-f002:**
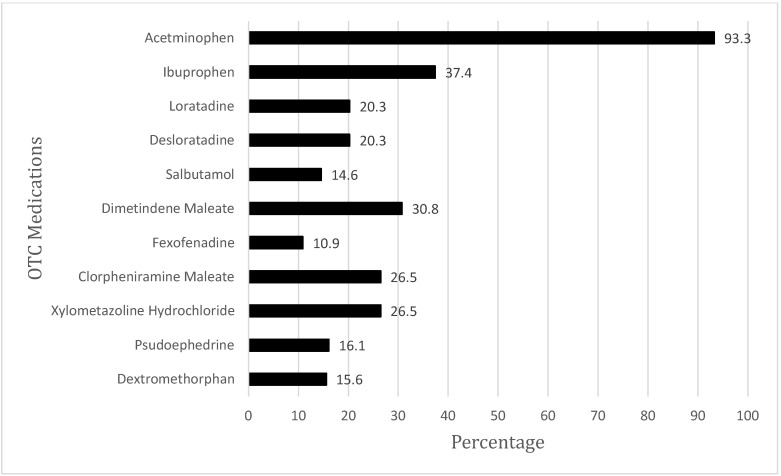
OTC medications that are commonly used by parents.

**Table 1 ijerph-20-01193-t001:** Sociodemographic characteristics of the participants.

	Numbers N = 211	Percentage %
Gender		
Male	96	45.5
Female	115	54.5
Nationality		
Saudi	191	90.5
Non-Saudi	20	9.5
Age (years)		
18–30	80	37.9
31–40	73	34.6
41–50	36	17.1
>50	22	10.4
Educational Level		
Basic *	30	14.3
Undergraduate	154	72.9
Postgraduate	27	12.8
Marital Status		
Married	182	86.7
Not Married (divorced)	29	13.3
Employment Status		
Employed	128	60.7
Unemployed	66	31.2
Retired	17	8.1
Monthly Income of Parents (SAR)		
1000–5000	42	19.9
>5000–10,000	74	35.1
>10,000	95	45.0

* Basic education (comprises elementary, intermediate, and high school education).

**Table 2 ijerph-20-01193-t002:** The relationship between [1] sociodemographic characteristics and [2] OTC drug consumption in children. Statistical significance was determined at a *p*-value of <0.005.

		OTC Drug Consumption in Children					
		No		Yes		Total	*p* Value
		N	%	N	%		
Gender	Male	29	30.2	67	69.8	96	0.027
	Female	40	34.8	75	65.2	115	
Nationality	Saudi	54	28.3	137	71.7	191	0.025
	Non-Saudi	15	75	5	25	20	
Age (years)	18–30	24	30	56	70	80	0.027
	31–40	27	37.0	46	63.0	73	
	41–50	14	38.9	22	61.1	36	
	>50	4	18.2	18	81.8	22	
Educational Level	Basic	17	56.0	13	44.0	30	<0.001
	Undergraduate	43	27.9	111	72.1	154	
	Postgraduate	9	33.3	18	66.7	27	
Marital Status	Married	65	35.7	117	64.3	182	<0.001
	Not Married (divorced)	4	13.8	25	86.2	29	
Employment Status	Employed	40	31.3	88	68.8	128	<0.001
	Unemployed	22	33.3	44	66.7	66	
	Retired	7	41.2	10	58.8	17	
Monthly Income of Parents (SAR)	1000–5000	18	42.9	24	57.1	42	0.004
	>5000–10,000	29	39.2	45	60.8	74	
	>10,000	22	23.2	73	76.8	95	

**Table 3 ijerph-20-01193-t003:** Side effects of OTC medications and parents’ response when side effects of OTC medications appear in their children.

	N	%
Side effects of OTC medications		
Urinary retention	3	1.5
Dry mouth	30	14.2
Itchy skin	36	17.0
Runny nose	43	20.3
Shortness of breath	16	7.5
Restlessness	20	9.5
Drowsiness	56	26.5
Abdominal pain	20	9.5
Nausea	24	11.3
Vomiting	30	14.2
Constipation	14	6.6
Diarrhoea	35	16.5
Loss of appetite	45	21.3
Parents’ response to side effects		
Stop giving OTC medicine	71	33.6
Keep giving the same OTC medicine	3	1.4
Use another OTC medicine	21	10.0
Visit a family physician	110	52.1
Visit the community pharmacy	6	2.8

## Data Availability

Data available upon request.

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
