# Peer review of "Over-the-Counter Medication Use among Parents in Saudi Arabia"

_ijerph, 2023, doi:10.3390/ijerph20021193_

Round 1
Reviewer 1 Report
Thank you for conducting this research! It is important to get a lay of the land so that you can make changes to better the health of all patients. Please find my general comments below
General:
1. You state "better perspective on OTC medication use among parents" - may I recommend that you make this a bit more specific because at the moment it is a bit too generic, maybe mention what you are specifically looking at
Abstract
1. Line 25: I am not sure how to understand the relevance of this sentence "While females represented 54.5% of the participants included in the study, parents belonging to the 18-to- 26 30-year-old group comprised the highest percentage (37.9%) and most of the parents (72.9%) had received an undergraduate education."
2. Line 32: I do not understand this sentence "The relationship between sociodemographic characteristics and educational level, marital status, and employment status was significant (p < 0.001)" --> what specifically is significant?
Introduction
1. Line 43: "In the main..." I am not familiar with this phrasing
2. Line 74: "The study aims to gain a better perspective on OTC medication use" - please see my statement regarding this wording above
Methods and Materials
1. How did you identify / find participants?
2. Which social media platforms did you use and why did you select these? How did you know which parents to include / exclude?
3. What are the serious medical conditions that excluded participants from the study and how did you know what these were if the participants were identified from social media?
4. Was the data that you evaluated already de-identified?
5. You never included what the actual sample size was supposed to be
Results
1. What is the reason behind looking at gender if you were investigating the perspectives of parents?
2. Line 119 - 123: I do not understand the correlation - was it associated with increased OTC use of less OTC use?
3. Table 2 is very difficult to understand, I am not sure what conclusion you are trying to reach with this data
Conclusion
Maybe you can include your thoughts about what additional research should be done in this field now that you have this kind of data.
Author Response
General:
- You state "better perspective on OTC medication use among parents" - may I recommend that you make this a bit more specific because at the moment it is a bit too generic, maybe mention what you are specifically looking at
Thank you for your valuable feedback
- Done
- Changed to “this study aims to investigate the attitude and knowledge among parents towards OTC medication use in the Jeddah region, Saudi Arabia”
Abstract
- Line 25: I am not sure how to understand the relevance of this sentence "While females represented 54.5% of the participants included in the study, parents belonging to the 18-to- 26 30-year-old group comprised the highest percentage (37.9%) and most of the parents (72.9%) had received an undergraduate education."
- Done
- Sorry for the grammar mistake - Changed to “Females represented 54.5% of the participants included in the study. Parents belonging to the 18-to-30-year-old group comprised the highest percentage (37.9%), and most of the parents (72.9%) had received an undergraduate education”
- Line 32: I do not understand this sentence "The relationship between sociodemographic characteristics and educational level, marital status, and employment status was significant (p < 0.001)" --> what specifically is significant?
- Done
- Rephrased- The relationship between the sociodemographic characteristics (including educational level, marital status, and employment status) and OTC drug consumption was significant (p<0.001).
Introduction
- Line 43: "In the main..." I am not familiar with this phrasing
- Done
- Changed to “mainly”
- Line 74: "The study aims to gain a better perspective on OTC medication use" - please see my statement regarding this wording above
- Done
- Changed to “this study aims to investigate the attitude and knowledge among parents towards OTC medication use in the Jeddah region, Saudi Arabia”
Methods and Materials
- How did you identify / find participants?
- Done
- Explained - The participants were randomly approached by sending the electronic questionnaire through social media (including Twitter and WhatsApp) and Google Forms to download the responses. We attached a note in Arabic to the questionnaire saying (if you are a parent with young children, you can join the study. However, you cannot participate if you cannot provide informed consent or if your child has serious medical or surgical conditions requiring hospitalisation ).
- Which social media platforms did you use and why did you select these? How did you know which parents to include / exclude?
- Done
- Explained- Social media (including Twitter and WhatsApp) were used to send the questionnaire, and Google Forms to download the responses. We attached a note in Arabic to the questionnaire saying (if you are a parent with young children, you can join the study. However, you cannot participate if you cannot provide informed consent or if your child has serious medical or surgical conditions requiring hospitalisation ).
- What are the serious medical conditions that excluded participants from the study and how did you know what these were if the participants were identified from social media?
- Done
- Explained - Any serious medical or surgical conditions requiring hospitalisation.We attached a note in Arabic to the questionnaire saying (if you are a parent with young children, you can join the study. However, you cannot participate if you cannot provide informed consent or if your child has serious medical or surgical conditions requiring hospitalisation ).
- Was the data that you evaluated already de-identified?
- Done – Yes, all personally identifiable information has been removed
- You never included what the actual sample size was supposed to be
- Done
- Explained - The sample size was calculated using Slovin’s formula, with a population size of 109 participants from a recently published study in Qassim region, Saudi Arabia by Al-Ghofaili et al. 2021, with a confidence interval (CI) of 95% and a margin of error of 5% (9).
Results
- What is the reason behind looking at gender if you were investigating the perspectives of parents?
- We wanted to see who could answer the questionnaire and know about OTC use in children
- Line 119 - 123: I do not understand the correlation - was it associated with increased OTC use of less OTC use?
- Done
- Rephrased- There was a dependent relationship between sociodemographic characteristics (including educational level, marital status, and employment status) and OTC drug consumption in children (p<0.001), as shown in Table 2.
- Table 2 is very difficult to understand, I am not sure what conclusion you are trying to reach with this data
- Done
- Rephrased-There was a dependent relationship between sociodemographic characteristics (including educational level, marital status, and employment status) and OTC drug consumption in children (p<0.001), as shown in Table 2. However, sociodemographic characteristics (including gender, nationality, and age ) were not related to OTC drug consumption in children (p=0.027), (p=0.025), (p=0.027), respectively.
Conclusion
Maybe you can include your thoughts about what additional research should be done in this field now that you have this kind of data.
- Done
- Added- Additional research is needed to monitor OTC use among parents and identify factors contributing to self-medication to provide successful strategies to improve medication practices in Saudi Arabia.
Reviewer 2 Report
The purpose of this study is to examine the current status of over-the-counter medications in Saudi Arabia. The topic of this manuscript fits with the journal scope. Several suggestions are provided for the authors’ reference.
1. Materials and Methods
(1) Study Design: How the questionnaires were randomly distributed to respondents should be clearly explained.
2. Results
(1) It is suggested to explain with some examples for why 39 responses were incomplete and removed from subsequent analysis.
(2) The statistical figures in Table 1 and Table 2 are inconsistent. For example, the number of not married is 28 in Table 1 but 29 in Table 2. The number of ‘Employment status’ has the same problem. And Chi-square test has to be re-analyzed in Table 2.
(3) P. 5, line 127: The number of 37.3 is suggested to be rounded as the same with the number (37.4) in Figure 1.
3. Discussion: Both age and marital status have significant relationships with OTC drug consumption in children (Based on Table 2). These findings should be discussed in this section.
Author Response
The purpose of this study is to examine the current status of over-the-counter medications in Saudi Arabia. The topic of this manuscript fits with the journal scope. Several suggestions are provided for the authors’ reference.
- Materials and Methods
- Study Design: How the questionnaires were randomly distributed to respondents should be clearly explained.
Thank you for your valuable feedback
- Done
- Explained -The participants were randomly approached by sending the electronic questionnaire through Social media (including Twitter and WhatsApp) and Google Forms to download the responses.. We attached a note in Arabic to the questionnaire saying (if you are a parent with young children, you can join the study. However, you cannot participate if you cannot provide informed consent or if your child has serious medical or surgical conditions requiring hospitalisation ).
- Results
(1) It is suggested to explain with some examples for why 39 responses were incomplete and removed from subsequent analysis.
- Done
- Changed to- Participants who did not answer all 13 questions from the three-part questionnaire
- The statistical figures in Table 1 and Table 2 are inconsistent. For example, the number of not married is 28 in Table 1 but 29 in Table 2. The number of ‘Employment status’ has the same problem. And Chi-square test has to be re-analyzed in Table 2.
- Done – Sorry for the typing mistakes
(3) P. 5, line 127: The number of 37.3 is suggested to be rounded as the same with the number (37.4) in Figure 1.
- Done
- Corrected to 37.4
- Discussion: Both age and marital status have significant relationships with OTC drug consumption in children (Based on Table 2). These findings should be discussed in this section.
- Done
- Added – age was 0.027
- We also found a dependent relation (p<0.001) between marital status and self-medication with OTCs and between employment status and self-medication with OTCs. Several studies conducted in high-income countries have concluded that marriage is positively related to high income, and those with a higher income have a greater tendency to use OTC medications (24,25).